# Diagnostic Roles of Immunohistochemistry in Thymic Tumors: Differentiation between Thymic Carcinoma and Thymoma

**DOI:** 10.3390/diagnostics10070460

**Published:** 2020-07-06

**Authors:** Jae-Han Jeong, Jung-Soo Pyo, Nae-Yu Kim, Dong-Wook Kang

**Affiliations:** Department of Thoracic and Cardiovascular Surgery, Chosun University Hospital, Chosun University School of Medicine, Gwangju 61453, Korea; thoracic_surgeon@chosun.ac.kr; Department of Pathology, Daejeon Eulji University Hospial, Eulji University School of Medicine, Daejeon 35233, Korea; jspyo@eulji.ac.kr; Department of Internal Medicine, Daejeon Eulji University Hospital, Eulji University School of Medicine, Daejeon 35233, Korea; naeyu46@eulji.ac.kr; Department of Pathology, Chungnam National University Sejong Hospital, 20 Bodeum 7-ro, Sejong 30099, Korea; Department of Pathology, Chungnam National University School of Medicine, 266 Munhwa Street, Daejeon 35015, Korea

**Keywords:** thymus, thymic carcinoma, thymoma, immunohistochemistry, meta-analysis, diagnostic test accuracy review

## Abstract

*Background:* The present study aims to evaluate the diagnostic roles of various immunohistochemical (IHC) markers in thymic tumors, including thymic carcinoma (TC) and thymoma (TM). *Methods:* Eligible studies were obtained by searching the PubMed databases and screening the searched articles. Thirty-eight articles were used in the present meta-analysis and included 636 TCs and 1861 TMs. Besides, for IHC markers with statistical significance, a diagnostic test accuracy review was performed. *Results:* The comparison of various IHC expressions between TC and TM was performed for 32 IHC markers. Among these IHC markers, there were significant differences between TC and TM for beta-5t, B-cell lymphoma 2 (Bcl-2), calretinin, CD1a, CD5, carcinoembryonic antigen (CEA), cytokeratin19 (CK19), CD117, glucose transporter 1 (Glut-1), insulin-like growth factor 1 receptor (IGF-1R), mesothelin, MOC31, mucin1 (MUC1), p21, and terminal deoxynucleotidyl transferase (TdT). Markers with higher expressions in TCs were Bcl-2, calretinin, CD5, CEA, CD117, Glut-1, IGF-1R, mesothelin, MOC31, MUC1, and p21. Among these markers, there were no significant differences between TC and TM type B3 in immunohistochemistries for Bcl-2 and CK19. On the other hand, *β*-catenin and CD205 showed a considerable difference in IHC expressions between TC and TM type B3, but not between TC and overall TM. In diagnostic test accuracy review, MUC1 and beta-5t were the most useful markers for TC and TM, respectively. *Conclusions*: Taken together, our results showed that the expression rates for various IHC markers significantly differed between TC and TM. The IHC panel can be useful for differentiation from limited biopsied specimens in daily practice.

## 1. Introduction

Thymic epithelial tumors (TETs), which originate from thymic epithelial cells, include thymoma (TM), thymic carcinoma (TC), and thymic neuroendocrine tumors [1,2]. These TETs have different biological functions, histological findings, and genomic profiles [2,3]. TMs can be classified into type A, AB, B1, B2, B3, and C (TC) based on the World Health Organization (WHO) classification [1]. According to the WHO classification, TETs, regardless of subtype or histology, are classified as malignant tumors [1,4]. The incidence of TC is approximately 22% [5], with squamous cell carcinoma being the most common subtype of TC, accounting for approximately 70% of all TCs [1,6]; accurate differentiation between thymic squamous cell carcinoma and TM type B3 is required [1]. Further, focal squamous differentiation and keratinization can be found in other types of TC, such as lymphoepithelioma-like carcinoma and basaloid carcinoma [1]. In the diagnosis of anterior mediastinal tumors, the direct invasion and metastasis of pulmonary squamous cell carcinoma are required to differentiate between these tumors and TC. The pathologic diagnosis of an anterior mediastinal tumor is essential when determining treatment modality and prognosis [2]. TC patients generally show a higher stage and worse prognosis compared to TM patients [3,7], but the differential diagnosis is challenging when the given specimen is a small biopsied tissue. Previous studies reported the diagnostic implications of various immunohistochemical (IHC) markers, including CD117, a marker closely related to the invasion and metastasis of tumor cells, which is highly expressed in TC [2]. In addition, Nakagawa et al. reported that the combination of CD117 and CD5 was useful for differentiating between thymic and lung squamous cell carcinoma [8]. Insulin-like growth factor-1 receptor (IGF-1R) was proposed as a potential therapeutic target for TET and is more frequently expressed in TC than in TM [9,10,11].

Although various IHC expressions were reported, single specific markers for each TET are not available. In our study, we evaluated the IHC expression patterns of TCs and TMs, alongside the performance of a diagnostic test accuracy review for various IHC markers.

## 2. Materials and Methods

### 2.1. Published Study Search and Selection Criteria

Relevant articles were obtained by searching the PubMed database through 31 January 2020. For searching, the keywords used were as follows: “thymic carcinoma or thymoma” and “immunohistochemistry”. The titles and abstracts of all searched articles were screened for inclusion and exclusion. Included articles had the information for the immunohistochemistry of the TC and TM. However, case reports, non-original articles, or those written in English were excluded from the present study. The PRISMA checklist shown in the Appendix A.

### 2.2. Data Extraction

Data associated with various IHC expressions of TC and TM were extracted from each of the eligible studies [2,8,9,10,11,12,13,14,15,16,17,18,19,20,21,22,23,24,25,26,27,28,29,30,31,32,33,34,35,36,37,38,39,40,41,42,43,44]. Two independent authors extracted all of the data. Extracted data were the author’s information, study location, number of patients analyzed, and tumor subtypes of TM. In addition, the expression rates by IHC markers were investigated in TC and TM.

### 2.3. Statistical Analyses

A meta-analysis was performed using the Comprehensive Meta-Analysis software 2.0 package (Biostat, Englewood, NJ, USA). The expression rates of various IHC markers were investigated by dividing them into TC and TM markers. Comparisons of IHC expressions between TC and TM type B3 were also performed. Heterogeneity between the studies was checked using *Q* and *I^2^* statistics and expressed as *p*-values. Additionally, sensitivity analysis was conducted to assess the heterogeneity of eligible studies and the impact of each study on the combined effects. Due to the use of various evaluation criteria and tumor types in the eligible studies, a random-effect model rather than a fixed-effect model was determined to be more suitable for this meta-analysis. The Begg’s funnel plot and Egger’s test were performed to assess publication bias, with fail-safe N and trim-fill tests additionally used to confirm the degree of publication bias if found. The results were considered statistically significant at *p* < 0.05. The diagnostic test accuracy review of various IHC markers was performed using R software ver. 3.6.3 (R Foundation for Statistical Computing, Vienna, Austria). We calculated the pooled sensitivity, specificity and diagnostic odds ratio (OR) according to individual data collected from each eligible study. By plotting the sensitivity and 1-specificity of each study, the summary receiver operating characteristic curve (SROC) was able to be constructed with curve fitting performed via linear regression. Due to all of the data being heterogeneous, accuracy data were pooled by fitting the SROC and measuring the area under the curve (AUC).

## 3. Results

### 3.1. Selection and Characteristics of the Studies

In this study, 934 relevant articles were searched for on the PubMed database and reviewed for a meta-analysis. Of these, 409 articles had no or lack of sufficient information for a meta-analysis. In addition, 267 were excluded due to non-original articles. Among the remaining articles, 220 reports were excluded for the following reasons: articles in other diseases (*n* = 105), non-human studies (*n* = 81), and a language other than English (*n* = 34) (Figure 1). Finally, 38 eligible articles were selected and included for the meta-analysis (Table 1). These studies included 2497 patients, including TC (*n* = 636) and TM (*n* = 1861).

### 3.2. Comparison of Immunohistochemical Expressions between Thymic Carcinoma and Thymoma 

First, the significant differences in IHC expressions between TC and TM were investigated. There were significant differences in immunohistochemistry for beta-5t, B-cell lymphoma 2 (Bcl-2), calretinin, CD1a, CD5, carcinoembryonic antigen (CEA), cytokeratin19 (CK19), CD117, Glut-1, IGF-1R, mesothelin, MOC31, MUC1, p21, and TdT (Table 2). Among these markers, Bcl-2, calretinin, CD5, CEA, CD117, glucose transporter 1 (Glut-1), insulin-like growth factor 1 receptor (IGF-1R), mesothelin, MOC31, mucin1 (MUC1), and p21 showed significantly higher expressions in TC than in TM. On the other hand, TMs have shown higher expressions of beta-5t, CD1a, CK19, and terminal deoxynucleotidyl transferase (TdT) than TC. In comparison with TC, the significant differences in IHC expressions for beta-catenin and CD205 were found in TM type B3, but not in overall TM. Among markers with a significant difference, the estimated expression rates of CD205 were 0.650 (95% CI 0.461–0.801) and 0.958 (95% CI 0.757–0.994) in TC and TM type B3, respectively (Table 3). In addition, the estimated expression rates of Glut-1 and IGF-1R were significantly higher in TC than in TM type B3. However, the Glut-1 and IGF-1R expression rates of TM type B3 were 52.6% and 64.6%, respectively.

### 3.3. Diagnostic Test Accuracy Review for Immunohistochemical Markers

The diagnostic test accuracy reviews were performed for candidates of IHC markers, which were showed the statistical differences between TC and TM type B3. Five positive markers, including CD5, CD117, Glut-1, and IGF-1R, MUC1, and four negative markers, including beta-5t, CD1a, CD205, and TdT, were included in the present analysis (Table 4). Among these markers, the most effective positive and negative markers may be MUC1 and beta-5t, 0.932 (95% CI 0.686–0.988), 0.847 (95% CI 0.505–0.968), 46.251 ( 95% CI 11.634–183.877), 0.921 and 1.000 ( 95% CI 0.927–1.000), 1.000 (95% CI 0.942–1.000), 571.396 (95% CI 33.356–9788.053), 0.985), in sensitivity, specificity, diagnostic OR, and AUC on SROC, respectively; Table 4. The orders of AUC on SROC were MUC1, Glut-1, CD117, IGF-1R, and CD5 in positive markers and were beta-5t, TdT, CD1a, and CD205 in negative markers.

## 4. Discussion

Although the prognosis of each subtype of TETs is not clear, the prognosis of Type B3 TMs is clearly different from other subtypes of TMs and TC [22,45,46]; however, histological similarities are often found in thymic squamous cell carcinoma and TM type B3 [4,22,45]. Due to the potential importance of differentiation between TC and TM type B3, various IHC markers, such as CD117 and CD5, were introduced and studied [1,4,8,12,30,47,48]. However, the accuracy of using these markers as a diagnostic test was not clarified [22,45,46]. This study presented the first meta-analysis and diagnostic test accuracy review of the diagnostic roles of various IHC markers in TETs, including TC and TM.

TETs can be differentiated by histological characteristics [1]; however, similar histologic findings are present between the subtypes of TET. Because each subtype exhibits different clinical behaviors and outcomes, a precise diagnosis is essential [14,49]. The treatment of choice for TETs is surgical resection, where possible [50]. In inoperable cases, the preoperative diagnosis may be more important in regard to treatment decisions, such as chemotherapy [51]. However, other malignant and benign tumors can also occur in the thymus. The diagnostic goals for biopsy specimens could lie in defining malignant or benign tumors and differentiating between TETs and others. However, specimens obtained via needle biopsy have some limitations in regard to histological diagnosis. Immunohistochemistry is useful for the diagnosis of small biopsied specimens. In addition, information regarding various protein expression patterns is useful in regard to understanding tumorigenesis and developing targeted drugs.

Various tumors can occur in the anterior mediastinum, including TETs, germ cell tumors, and metastatic tumors. Differentiating between TET subtypes can be useful when deciding on treatments and predicting prognoses. Due to the similar histological findings of TC and TM type B3, diagnoses from small biopsied specimens are challenging in daily practice. However, single and specific IHC markers for each tumor are not yet defined. Specific markers have high expression in the target and low expression in the reference. According to our results, positive markers may be suitable for CD5, CD117, MUC1, and Glut-1 in differentiating TC from TM type B3. On the other hand, beta-5t, CD1a, and TdT are considered to be negative markers. In this study, we initially evaluated the statistical differences between TC and TM using odds ratios. In addition, we analyzed the expression rates of each IHC marker with statistical significance between TC and TM. These results should be considered before applying diagnostic markers in daily practice. Diagnostic implications, regardless of the statistical significance, are limited when IHC expression rates are high in both compared subgroups. Therefore, our results showed that the estimated expression rates were useful for the selection of an IHC panel.

Kim et al. suggested that an IHC panel using EZH2, CD117, and CD205 was useful for differentiation between TC and TM type B3 [46]. In the previous study, EZH2 showed higher sensitivity (88.9%) and specificity (100%) in differentiating between TC and TM type B3 [46]. However, because the raw data for EZH2 immunohistochemistry were not shown, the present study could not analyze these results. Based on our results, CD205 exhibited significantly lower expression in TC than in TM type B3. However, the CD205 expression rates of TC and TM type B3 were 65.0% and 95.8%, respectively. This difference in expression rate was not useful for differentiation between two tumor groups; further, information regarding negative markers was not shown [46]. Some markers, such as beta-5t, CD1a, and TdT, showed higher expressions in TM type B3 than in TC, which may present negative marker candidates for TC. Therefore, an IHC panel using positive and negative markers could be useful for the diagnosis of TC and TM, and an IHC panel consisting of positive and negative markers for the diagnosis of an anterior mediastinal mass is recommended. This diagnostic test accuracy review was performed for IHC markers with significant differences in expression between TC and TM type B3. Positive markers included CD5, CD117, Glut-1, IGF-1R, MUC1 for TC, beta-5t, CD1a, CD205, and TdT for TM type B3. Markers with high sensitivity, specificity, and diagnostic odds ratios were considered to be more effective. Among the markers highly expressed in TC, Glut-1 showed the highest expression rate, 95.2%. However, in TM type B3, the expression rate of Glut-1 was 52.6% (95% CI 29.6–74.5%). In the diagnostic test accuracy review, Glut-1 showed the highest diagnostic odds ratio (46.251, 95% CI 11.634–183.877). MUC1 expression in TM type B3 ranged from 0% to 77.8% [13,21,36], indicating a limited diagnostic role of MUC1, as well as Glut-1. Based on these criteria, the important markers were shown to be CD5 and CD117 (positive) and TdT and beta-5t (negative) for TCs.

The direct invasion and metastasis of primary lung cancers in the anterior mediastinum can be detected [1]. Based on the histological findings, it is difficult to differentiate between lung and thymic squamous cell carcinomas. Among the markers expressed in TC, CD5, CD117, and CD205 are uncommonly expressed in primary lung cancers [8], thereby presenting these markers as possible candidates in the IHC panel of thymic tumors when differentiating primary lung cancers. According to our results, CD205 may be comparable in differentiation from tumors of lung origin because it is expressed in both TC and TM. In addition, CD205 is useful for defining thymic origin. Taken together, an IHC panel containing CD205 as a positive and a negative marker would be more effective.

This study has a limitation in that we performed the diagnostic test accuracy review for individual IHC markers, but it was difficult to conduct the diagnostic test accuracy review for a combination of markers due to limited eligible study information. Therefore, the recommended IHC panels could not clarify the diagnostic role of the differentiation of thymic masses.

## 5. Conclusions

In conclusion, our results showed that significant differences in IHC expression between TC and TM identified positive markers, including CD5, CD117, Glut-1, IGF-1R, and MUC1, and negative markers, including beta-5t, CD1a, CD205, and TdT against TC. An IHC panel including positive and negative markers, as well as CD205, could be useful to differentiate between thymic masses in daily practice.

## Figures and Tables

**Figure 1 diagnostics-10-00460-f001:**
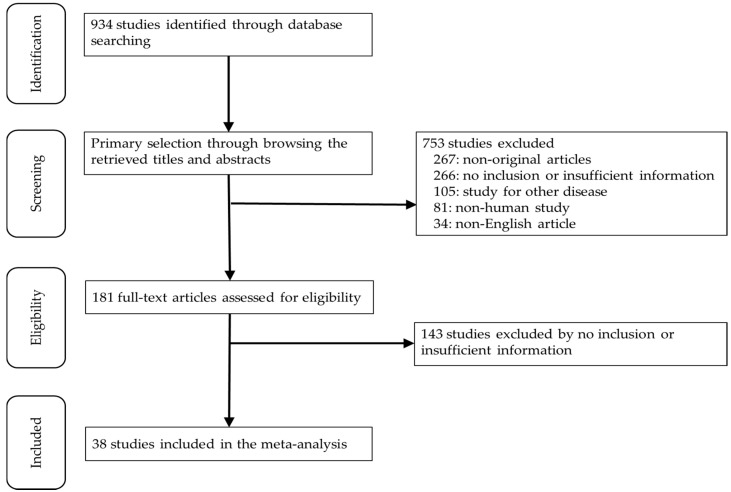
Flow chart of the searching strategy.

**Table 1 diagnostics-10-00460-t001:** Main characteristics of the eligible studies.

Study	Location	Number of Patients						
		Thymic Carcinoma	Thymoma	Type A	Type AB	Type B1	Type B2	Type B3
Adam 2014 [9]	Germany	24	45					
Chen 1996 [10]	Taiwan	26	15					
Cui 2011 [11]	China	4	39	2	5	7	14	11
Dorfman 1997 [12]	USA	24	41					
Du 2016 [13]	China	22	21					21
Girard 2009 [14]	USA	7	38	8			22	8
Girard 2010 [15]	USA	7	56	5	12	8	21	10
Hayashi 2013 [16]	Japan	18	17					17
Henley 2002 [17]	USA	6	36					
Hino 1997 [18]	Japan	19	17					
Hirabayashi 1997 [19]	Japan	4	36					
Hiroshima 2002 [20]	Japan	10	36	8	8	7	7	6
Kaira 2011 [21]	Japan	17	5					5
Khoury 2010 [22]	USA	12	54					17
Kornstein 1997 [23]	USA	24	85					
Laury 2011 [24]	USA	5	9					
Lee 2019 [25]	Korea	30	110	11	31	28	16	19
Mimae 2011 [26]	Japan	37	103					6
Mimae 2012 [27]	USA	37	103					6
Nakagawa 2005 [8]	Japan	20	50	10	10	10	10	10
Nonaka 2007 [28]	USA	16	58	9	19	7	16	7
Omatsu 2012 [29]	Japan	22	22	1	1	7	7	6
Pan 2003 [30]	Taiwan	22	35	9	10	4	7	5
Petrini 2010 [31]	Italy	13	105					
Remon 2017 [32]	France	12	84	4	25	8	27	20
Rieker 2006 [33]	Germany	4	30	8	6	5	6	5
Song 2012 [34]	China	15	87	3	29	5	22	28
Stefanaki 1997 [35]	Greece	2	29					
Su 2015 [36]	China	20	16					16
Suzuki 2018 [37]	Japan	10	7					
Tateyama 1999 [38]	Japan	7	18					
Thomas 2016 [39]	USA	34	29					
Thomas de Montpréville 2015 [40]	France	16	75	5	17	11	25	17
Tsuchida 2008 [41]	Japan	17	20		5	4	6	5
Weissferdt 2011 [42]	USA	31	60	30				
Wu 2019 [2]	China	22	128	11	35	19	40	23
Yamada 2011 [43]	Japan	13	41	3	17	7	10	4
Zucali 2010 [44]	Italy	8	101	15	28	24	8	24

**Table 2 diagnostics-10-00460-t002:** Meta-analysis for the odds ratio of various immunohistochemical expressions between thymic carcinoma and thymoma.

Marker	Number of Subsets	Fixed Effect [95% CI]	Heterogeneity Test [*p*-Value]	Random Effect [95% CI]	Egger’s Test [*p*-Value]
Androgen receptor	3	0.362 [0.120, 1.091]	0.063	0.740 [0.065, 8.450]	0.480
beta-5t	2	0.002 [0.000, 0.030]	0.564	0.002 [0.000, 0.030]	-
beta-catenin	2	0.829 [0.254, 2.704]	0.022	0.512 [0.027, 9.722]	-
Bcl-2	4	2.461 [1.043, 5.807]	0.637	2.461 [1.043, 5.807]	0.871
Calretinin	1	19.429 [2.218, 170.165]	1.000	19.429 [2.218, 170.165]	-
CD15	2	4.139 [1.413, 12.127]	0.022	2.263 [0.130, 39.382]	-
CD1a	2	0.052 [0.012, 0.223]	0.073	0.028 [0.001, 0.623]	-
CD205	2	0.221 [0.064, 0.759]	0.019	0.137 [0.006, 3.046]	-
CD5	11	52.560 [26.424, 104.547]	0.972	52.560 [26.424, 104.547]	0.034
CEA	2	45.273 [5.567, 368.160]	0.505	45.273 [5.567, 368.160]	-
CK19	2	0.061 [0.016, 0.224]	0.364	0.061 [0.016, 0.224]	-
CK5/6	4	0.191 [0.080, 0.459]	0.022	0.294 [0.054, 1.607]	0.283
c-Kit	12	41.444 [23.767, 72.267]	0.771	41.444 [23.767, 72.267]	0.024
Cyclin D1	2	0.407 [0.128, 1.298]	0.006	1.140 [0.022, 58.476]	-
E-cadherin	3	0.340 [0.170, 0.680]	0.001	0.400 [0.064, 2.516]	0.167
EGFR	6	0.311 [0.130, 0.741]	0.014	0.314 [0.066, 1.493]	0.964
Estrogen receptor	1	0.319 [0.012, 8.254]	1.000	0.319 [0.012, 8.254]	-
Glut-1	4	11.607 [3.003, 44.862]	0.100	15.187 [2.082, 110.780]	0.019
HBME	2	2.776 [0.337, 22.853]	0.088	2.763 [0.076, 100.781]	-
IGF-1R	6	10.216 [5.611, 18.602]	0.005	9.465 [2.869, 31.221]	0.806
Mesothelin	3	39.842 [12.067, 131.542]	0.876	39.842 [12.067, 131.542]	0.386
MOC31	2	18.019 [4.366, 75.113]	0.874	18.019 [4.366, 75.113]	-
MUC1	3	44.866 [11.273, 178.576]	0.786	44.866 [11.273, 178.576]	0.249
p21	2	10.270 [2.862, 36.849]	0.716	10.270 [2.862, 36.849]	-
p53	7	2.554 [1.077, 6.055]	0.029	3.199 [0.759, 13.481]	0.487
p63	3	0.239 [0.094, 0.610]	0.013	0.264 [0.028, 2.482]	0.924
PAX8	3	0.371 [0.107, 1.288]	0.065	0.539 [0.058, 4.989]	0.505
Progesterone receptor	2	1.681 [0.170, 16.597]	0.597	1.681 [0.170, 16.597]	-
Survivin	2	1.251 [0.358, 4.378]	0.103	0.733 [0.056, 9.558]	-
TdT	2	0.015 [0.003, 0.085]	0.206	0.014 [0.001, 0.126]	-
Thrombomodulin	1	0.449 [0.023, 8.896]	1.000	0.449 [0.023, 8.896]	-
WT-1	1	4.953 [0.193, 127.130]	1.000	4.953 [0.193, 127.130]	-

CI, confidence interval.

**Table 3 diagnostics-10-00460-t003:** Meta-analysis for the odds ratio of various immunohistochemical expressions between thymic carcinoma and thymoma type B3.

Marker	Type	Number of Subsets	Fixed Effect [95% CI]	Heterogeneity Test [*p*-Value]	Random Effect [95% CI]	Egger’s Test [*p*-Value]
beta-5t	Thymic carcinoma	2	0.031 [0.004, 0.188]	0.877	0.031 [0.004, 0.188]	-
	Thymoma type B3	2	0.948 [0.706, 0.993]	0.511	0.948 [0.706, 0.993]	-
beta-catenin	Thymic carcinoma	1	0.750 [0.448, 0.917]	1.000	0.750 [0.448, 0.917]	-
	Thymoma type B3	1	0.118 [0.030, 0.368]	1.000	0.118 [0.030, 0.368]	-
CD1a	Thymic carcinoma	2	0.127 [0.036, 0.360]	0.155	0.096 [0.012, 0.489]	-
	Thymoma type B3	2	0.847 [0.680, 0.935]	0.682	0.847 [0.680, 0.935]	-
CD205	Thymic carcinoma	2	0.650 [0.461, 0.801]	0.371	0.650 [0.461, 0.801]	-
	Thymoma type B3	2	0.958 [0.757, 0.994]	0.679	0.958 [0.757, 0.994]	-
CD5	Thymic carcinoma	5	0.722 [0.610, 0.812]	0.678	0.722 [0.610, 0.812]	0.318
	Thymoma type B3	5	0.100 [0.039, 0.233]	0.358	0.096 [0.035, 0.236]	0.110
CEA	Thymic carcinoma	1	0.750 [0.522, 0.892]	1.000	0.750 [0.522, 0.892]	-
	Thymoma type B3	1	0.029 [0.002, 0.336]	1.000	0.029 [0.002, 0.336]	-
c-Kit	Thymic carcinoma	11	0.688 [0.607, 0.759]	0.142	0.692 [0.591, 0.778]	0.532
	Thymoma type B3	11	0.099 [0.060, 0.160]	0.944	0.099 [0.060, 0.160]	0.005
Glut-1	Thymic carcinoma	4	0.952 [0.862, 0.985]	0.827	0.952 [0.862, 0.985]	0.017
	Thymoma type B3	4	0.495 [0.351, 0.640]	0.105	0.526 [0.296, 0.745]	0.621
IGF-1R	Thymic carcinoma	5	0.820 [0.720, 0.890]	0.580	0.820 [0.720, 0.890]	0.147
	Thymoma type B3	5	0.632 [0.495, 0.751]	0.179	0.646 [0.468, 0.791]	0.573
Mesothelin	Thymic carcinoma	1	0.417 [0.185, 0.692]	1.000	0.417 [0.185, 0.692]	-
	Thymoma type B3	1	0.028 [0.002, 0.322]	1.000	0.028 [0.002, 0.322]	-
MOC31	Thymic carcinoma	1	0.500 [0.244, 0.756]	1.000	0.500 [0.244, 0.756]	-
	Thymoma type B3	1	0.118 [0.030, 0.368]	1.000	0.118 [0.030, 0.368]	-
MUC1	Thymic carcinoma	3	0.849 [0.706, 0.930]	0.140	0.897 [0.666, 0.975]	0.034
	Thymoma type B3	3	0.270 [0.144, 0.449]	0.051	0.198 [0.048, 0.549]	0.462
p21	Thymic carcinoma	1	0.667 [0.376, 0.869]	1.000	0.667 [0.376, 0.869]	-
	Thymoma type B3	1	0.118 [0.030, 0.368]	1.000	0.118 [0.030, 0.368]	-
TdT	Thymic carcinoma	2	0.070 [0.018, 0.242]	0.611	0.070 [0.018, 0.242]	-
	Thymoma type B3	2	0.865 [0.690, 0.949]	0.336	0.865 [0.690, 0.949]	-

CI, confidence interval.

**Table 4 diagnostics-10-00460-t004:** Diagnostic test accuracy review of various immunohistochemical markers for differentiation between thymic carcinoma and thymoma type B3.

	Marker	Included Studies	Sensitivity (%) [95% CI]	Specificity (%) [95% CI]	Diagnostic OR [95% CI]	AUC on SROC
Thymic carcinoma	CD5	5	0.731 [0.622, 0.817]	0.967 [0.756, 0.996]	23.936 [7.693, 74.478]	0.725
	c-kit	11	0.709 [0.613, 0.790]	0.925 [0.873, 0.957]	23.623 [11.900, 46.894]	0.910
	Glut-1	4	0.942 [0.856, 0.978]	0.464 [0.225, 0.720]	11.823 [2.879, 48.549]	0.916
	IGF-1R	3	0.875 [0.760, 0.939]	0.250 [0.136, 0.415]	4.050 [1.087, 15.085]	0.758
	MUC1	3	0.932 [0.686, 0.988]	0.847 [0.505, 0.968]	46.251 [11.634, 183.877]	0.921
Thymoma type B3	beta-5t	2	1.000 [0.927, 1.000]	1.000 [0.942, 1.000]	571.396 [33.356, 9788.053]	0.985
	CD1a	2	0.743 [0.628, 0.832]	0.952 [0.504, 0.997]	35.919 [1.606, 803.371]	0.871
	CD205	2	1.000 [0.931, 1.000]	0.335 [0.165, 0.504]	11.735 [1.368, 100.632]	0.785
	TdT	2	0.879 [0.718, 0.954]	0.933 [0.769, 0.983]	93.458 [14.682, 594.912]	0.958

CI, confidence interval; OR, odds ratio; AUC, area under the curve; SROC, summary receiver operating characteristic.

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
