# Peer review of "Diagnostic Roles of Immunohistochemistry in Thymic Tumors: Differentiation between Thymic Carcinoma and Thymoma"

_diagnostics, 2020, doi:10.3390/diagnostics10070460_

Round 1
Reviewer 1 Report
I have reviewed the work of Jeong and coworkers, focused on performing a meta-analysis of usefulness of IHC markers for differentiating thymic tumors, namely thymomas from thymic carcinoma and the important distinction between thymoma B3 and thymic carcinoma. Better IHC markers are needed to aid Pathologists in this, especially in small biopsies, as the authors mention.
I have some comments and suggestions:
- written English should be improved, as certain sentences are a bit confusing and punctuation is missing;
- on section 2.1 authors do not mean that english articles were excluded - they mean the opposite
- can authors explain better how they assessed "expression" when doing the analyses of the publications? For instance, authors often refer "On the other hand, TMs have shown higher expressions of beta-5t..." - what was this higher expression? Higher intensity of staining, higher proportion of stained cells, or simply higher proportion of cases with evidence of staining?
- also, the articles used by the authors in their final analyses included only resection specimens or also biopsies? how do authors deal with this heterogeneity, since small biopsies have a lot less representation of tumor and its components, which can express a certain marker more evidently?
- the first paragraph of section 3.3 should be improved; some parts are not gramatically correct and the section with all the sensitivities, specificities and 95% CI is not easily read, so authors should find another way to refer to this
- the discussion is a bit repetitive if one reads it through; namely, authors repeat several times what is a "positive markers" or "negative marker". Being so repetitive the reader will lose the final take home message.
- authors in the end conclude on the greatest usefulness of specific markers, after having analyzing dozens of them: CD5, CD117, TdT and beta-5t. To my view these markers deserve comments on their biological role within these tumors. The discussion would benefit a lot from having some biological input, that triggers these biomarkers to be useful in the diagnosis.
- related to this last comment, authors could aim at proposing an algorithm, as a image, of how they envision that these biopsies or resection specimens should be approached, including the application of these biomarkers.
Author Response
We thank the reviewers for comments on our manuscript (ID: diagnostics-834226)
We tried to address the points raised by the reviewers as best as we can. The specific responses to the reviewers’ comments are described in Reply to Reviewers. We also fixed other unintended errors in our manuscript. Please see the attached file.
I have reviewed the work of Jeong and coworkers, focused on performing a meta-analysis of usefulness of IHC markers for differentiating thymic tumors, namely thymomas from thymic carcinoma and the important distinction between thymoma B3 and thymic carcinoma. Better IHC markers are needed to aid Pathologists in this, especially in small biopsies, as the authors mention.
I have some comments and suggestions:
written English should be improved, as certain sentences are a bit confusing and punctuation is missing;
Response: As suggested by a reviewer, We added some details to make the statement clearly and informative in the manuscript. Changes or corrected sentences were shown in blue color.
We underwent English editing by MDPI’s English editing service (ID: english-18107) for corrected use of grammar and standard medical terms. The certificate by MDPI’s English editing service is as bellow.
on section 2.1 authors do not mean that english articles were excluded - they mean the opposite
Response:
We corrected the comment for the exclusion criteria as below:
However, case reports, non-original articles, or those written in non-English were excluded from the present study.
can authors explain better how they assessed "expression" when doing the analyses of the publications? For instance, authors often refer "On the other hand, TMs have shown higher expressions of beta-5t..." - what was this higher expression? Higher intensity of staining, higher proportion of stained cells, or simply higher proportion of cases with evidence of staining?
Response:
The term “expression” means the positive rate. We corrected the terminology in the revised manuscript.
also, the articles used by the authors in their final analyses included only resection specimens or also biopsies? how do authors deal with this heterogeneity, since small biopsies have a lot less representation of tumor and its components, which can express a certain marker more evidently?
Response:
As pointed out, in daily practices, the pretreatment diagnosis may be made using needle biopsy samples. Basically, the pretreatment diagnosis using needle biopsy samples is based on the concept, which needle biopsy sample is representative of the tumor. This concept applies to immunohistochemistry as well as histologic examination.
Most of the eligible studies obtained the immunohistochemical results using surgical specimens. In addition, the results of the meta-analysis were analyzed using the random-effect model because of the heterogeneity of specimens.
In addition, we added the type of evaluated specimen in Table 1.
the first paragraph of section 3.3 should be improved; some parts are not gramatically correct and the section with all the sensitivities, specificities and 95% CI is not easily read, so authors should find another way to refer to this
Response:
We corrected the sentence as below:
Among these markers, the most effective positive and negative markers may be MUC1 (sensitivity: 0.932, 95% CI 0.686–0.988; specificity: 0.847, 95% CI 0.505-0.968; diagnostic OR: 46.251, 95% CI 11.634–183.877; AUC on SROC: 0.921) and beta-5t (sensitivity: 1.000, 95% CI 0.927–1.000; specificity: 1.000, 95% CI 0.942–1.000; diagnostic OR; 571.396, 95% CI 33.356–9788.053; AUC on SROC: 0.985; Table 4).
the discussion is a bit repetitive if one reads it through; namely, authors repeat several times what is a "positive markers" or "negative marker". Being so repetitive the reader will lose the final take home message.
Response:
As pointed out, the term “marker” was repeated in the manuscript. This paper is for diagnostic roles of immunohistochemistry marketers, so it has to be a bit repetitive. To increase readability, the use of the term was restricted and modified in the revised manuscript.
authors in the end conclude on the greatest usefulness of specific markers, after having analyzing dozens of them: CD5, CD117, TdT and beta-5t. To my view these markers deserve comments on their biological role within these tumors. The discussion would benefit a lot from having some biological input, that triggers these biomarkers to be useful in the diagnosis.
Response:
As a recommendation, we added the explanations in the revised manuscript as below:
Although CD5, CD117, and Glut-1 are not thymus-specific, these markers are useful markers for thymic carcinoma [43]. In thymoma, beta-5t, which is expressed exclusively in the thymic cortical epithelium, shows the immunoreactivity for neoplastic epithelial cells of the type B component [43]. TdT is specific for thymocytes, which are the lymphocytic component in a population of immature T lymphocytes [24].
related to this last comment, authors could aim at proposing an algorithm, as a image, of how they envision that these biopsies or resection specimens should be approached, including the application of these biomarkers.
Response:
Our results were obtained using a meta-analysis, and we got the results for individual markers. In addition, because there was no raw data of the individual study, no results were obtained by a combination of markers. In order to show diagnostic flow using the algorithm, a decision-tree analysis is appropriated. Our results showed the results for the diagnostic roles of positive and negative immunohistochemical markers. We previously described the limitation in the manuscript.

Reviewer 2 Report
Dear Authors,
The main value of this narrative review is the fact that it is addressing a topic of hot and wide interest in endocrinology. Despite some weakness found on design and structure of the text, I found that this article provides a significant useful overview.
1) Introduction is too short and requests more explanation what is/isn’t known/ is controversial about IHC diagnosis in each of the mentioned nosological form. Thus, I would kindly suggest to expand the info regarding this in this section.
2) Although in narrative reviews authors do not need to specify the search strategy, I would suggest in the 2.1 section to give more explanation about that how the comprehensive search of publications has been performed (eg: period of the search, years from what to what the papers were selected …etc.). It is highly not enough to mentioned that “Relevant articles were obtained by searching the PubMed database through January 31, 2020…”. I would also invite the authors to describe the number of citations found, as well as separate the “historical” papers (there are around 7 sources covering the previous century) in small section. As IHC has also changed over the years, it would be recommended to describe what was/wasn’t different/changed/why if relevant from the early IHC diagnostic time until the nowadays in pathomorphodiagnosis of TC and TM.
3) I would suggest authors to revise the Conclusions to avoid the extra words “In conclusion, our results showed…” and to make these final sentences more precise in such a way.
Author Response
We thank the reviewers for comments on our manuscript (ID: diagnostics-834226)
We tried to address the points raised by the reviewers as best as we can. The specific responses to the reviewers’ comments are described in Reply to Reviewers. We also fixed other unintended errors in our manuscript.
The main value of this narrative review is the fact that it is addressing a topic of hot and wide interest in endocrinology. Despite some weakness found on design and structure of the text, I found that this article provides a significant useful overview.
1) Introduction is too short and requests more explanation what is/isn’t known/ is controversial about IHC diagnosis in each of the mentioned nosological form. Thus, I would kindly suggest to expand the info regarding this in this section.
Response:
By a recommendation, we added the explanation in the revised manuscript.
2) Although in narrative reviews authors do not need to specify the search strategy, I would suggest in the 2.1 section to give more explanation about that how the comprehensive search of publications has been performed (eg: period of the search, years from what to what the papers were selected …etc.). It is highly not enough to mentioned that “Relevant articles were obtained by searching the PubMed database through January 31, 2020…”. I would also invite the authors to describe the number of citations found, as well as separate the “historical” papers (there are around 7 sources covering the previous century) in small section. As IHC has also changed over the years, it would be recommended to describe what was/wasn’t different/changed/why if relevant from the early IHC diagnostic time until the nowadays in pathomorphodiagnosis of TC and TM.
Response:
As pointed out, seven literatures are published before 2000. Among the literature searched, the literature that does not conform to the current diagnostic criteria was excluded. However, we did not set exclusion criteria at the time of study. Some studies have not used automated equipment even after 2000, so it is the study period that affects the staining method. However, because the effects of the timing of the study cannot be completely ruled out, the sensitivity analysis was performed. However, each study before 2000 confirmed that the impact on the estimated value was not significant.
3) I would suggest authors to revise the Conclusions to avoid the extra words “In conclusion, our results showed…” and to make these final sentences more precise in such a way.
Response:
As a recommendation, we corrected the conclusion in the revised manuscript.

Round 2
Reviewer 1 Report
No further comments
Reviewer 2 Report
Dear Authors,
The authors almost addressed all comments and the paper has been improved. Following second review, a few issues/clarifications remain. As I couldnt find anywhere in the text the full titles of the markers, I would like to ask the authors to decipher the abbreviations of the markers where they are mentioned in the first turn and at the Tables, too. For instance, Epidermal growth factor receptor 1 (EGFR-1), Insulin growth factor 1(IGF-1) etc. This is the last issue standing between you and my decision to move the paper for acceptance!